# Genetic Dissection of Growth and Eco-Physiological Traits Associated with Altitudinal Adaptation in Sakhalin Fir (*Abies sachalinensis*) Based on QTL Mapping

**DOI:** 10.3390/genes12081110

**Published:** 2021-07-22

**Authors:** Susumu Goto, Hideki Mori, Kentaro Uchiyama, Wataru Ishizuka, Haruhiko Taneda, Masaru Kono, Hiromi Kajiya-Kanegae, Hiroyoshi Iwata

**Affiliations:** 1The University of Tokyo Forests, Graduate School of Agricultural and Life Sciences, The University of Tokyo, Tokyo 113-8657, Japan; 2Department of Forest Molecular Genetics and Biotechnology, Forestry and Forest Products Research Institute, 1 Matsunosato, Tsukuba 305-8687, Ibaraki, Japan; morih@ffpri.affrc.go.jp (H.M.); kruchiyama@affrc.go.jp (K.U.); 3Forestry Research Institute, Hokkaido Research Organization, Koushunai, Bibai 079-0166, Hokkaido, Japan; wataru.ishi@gmail.com; 4Department of Biological Sciences, Graduate School of Science, The University of Tokyo, 7-3-1 Hongo, Bunkyo-ku, Tokyo 113-0033, Japan; taneda@bs.s.u-tokyo.ac.jp (H.T.); konom07@bs.s.u-tokyo.ac.jp (M.K.); 5Research Center for Agricultural Information Technology, NARO (RCAIT), 2-14-1 Nishi-Shimbashi, Minato-ku, Tokyo 105-0003, Japan; h.kanegae@affrc.go.jp; 6Laboratory of Biometry and Bioinformatics, Department of Agricultural and Environmental Biology, Graduate School of Agricultural and Life Sciences, The University of Tokyo, 1-1-1 Yayoi, Bunkyo-ku, Tokyo 113-8657, Japan; iwata@ut-biomet.org

**Keywords:** altitude, chlorophyll fluorescence, crown area, linkage map, local adaptation, needle morphology, phenology, RAD-seq, TodoFirGene

## Abstract

(1) Background: The genetic basis of local adaptation in conifers remains poorly understood because of limited research evidence and the lack of suitable genetic materials. Sakhalin fir (*Abies sachalinensis*) is an ideal organism for elucidating the genetic basis of local adaptation because its altitudinal adaptation has been demonstrated, and suitable materials for its linkage mapping are available. (2) Method: We constructed P336 and P236 linkage maps based on 486 and 516 single nucleotide polymorphisms, respectively, that were derived from double digest restriction site-associated DNA sequences. We measured the growth and eco-physiological traits associated with morphology, phenology, and photosynthesis, which are considered important drivers of altitudinal adaptation. (3) Results: The quantitative trait loci (QTLs) for growth traits, phenology, needle morphology, and photosynthetic traits were subsequently detected. Similar to previous studies on conifers, most traits were controlled by multiple QTLs with small or moderate effects. Notably, we detected that one QTL for the crown area might be a type-A response regulator, a nuclear protein responsible for the cytokinin-induced shoot elongation. (4) Conclusion: The QTLs detected in this study include potentially important genomic regions linked to altitudinal adaptation in Sakhalin fir.

## 1. Introduction

Local adaptation is an evolutionary process that is considered a home-site advantage, in which a genotype performs best in its native site [1]. As sessile organisms, plants are directly affected by the local climate, which is one of the most important drivers of adaptation [2]. However, a pairwise comparison between local and foreign plants revealed that local adaptation was observed in only 45% of the 1032 studied populations [3]. Hence, local adaptation is determined by the balance between gene flow and selection intensity [4]. For example, forest tree species with extensive gene flow generally exhibit local adaptation at a wide scale [5]. Selection drivers are usually complex because of a combination of climatic factors at distant sites, so the genetic basis of local adaptation in conifers is poorly understood.

Because the environmental conditions in an altitudinal gradient sharply change at a short spatial scale, these altitudinal gradients can be considered natural experiments [6]. High-altitude zones are generally characterized by a short growing season, high-intensity UV radiation, a low temperature, undeveloped soil, and heavy snow cover in combination with wind action [7]. Common garden data revealed that altitudinal clines for growth and bud phenology were present in conifers [8,9,10]. The clines from other common garden experiments also suggested the existence of local adaptation [11], but strict evidence was only obtained using reciprocal transplant experiments [12]. In herbaceous species, reciprocal transplant experiments at different altitudes demonstrated the local adaptation to native altitudinal zones [13,14]. However, for conifers, reciprocal transplant experiments at altitudinal gradients were rarely performed because of their large sizes and long generation times.

Elucidating the genetic basis of local adaptation is a hot topic in evolutionary biology [15]. Sakhalin fir (*Abies sachalinensis*) is an ideal organism for determining the genetic basis of altitudinal adaptation because the altitudinal clines for its growth and morphological traits were already revealed by common garden experiments [16] and provenance trials [17]. Reciprocal transplant experiments also demonstrated that the survival and growth rates of Sakhalin fir transplants were higher in planting sites close to their native altitudinal zones [18]. Additionally, segregated populations for altitudinal adaptation were already established for this species [19]. Using the segregated populations, several quantitative trait loci (QTLs) for phenology and growth traits have been identified in seedlings using linkage maps for this species [19]. Furthermore, the transcriptome database TodoFirGene (http://plantomics.mind.meiji.ac.jp/todomatsu/), containing 158,542 de novo transcriptome sequences in 69,618 loci, was readily available and previously assisted in determining the function of genes in Sakhalin fir [20].

Sakhalin fir is an ecologically and economically important boreal coniferous species distributed across Sakhalin Island, the Southern Kuril Islands, and Hokkaido, northern Japan [21]. In Hokkaido, the contrasting region-dependent climatic conditions affect the genetic components and structure of this species [22]. Provenance trials revealed that four climatic variables, namely winter solar radiation, warmth index, maximum snow depth, and spring solar radiation, are essential for the growth of Sakhalin fir [23]. At the regional scale, the home-site advantage of this species was also detected in regions with distinct climates [24].

In the present study, we aimed to detect the QTLs for growth and bud phenology and to identify the eco-physiological traits that are genetically controlled by QTLs. We first measured the growth and eco-physiological traits that were potentially associated with altitudinal adaptation in a segregated population of Sakhalin fir and subsequently conducted QTL analysis using linkage maps constructed with newly developed single nucleotide polymorphism (SNP) markers to identify the genomic regions linked with traits involved in altitudinal adaptation.

## 2. Materials and Methods

### 2.1. Plant Materials

To establish a mapping population, we performed reciprocal pollination between two hybrids (high- × low-altitude genotypes) derived from different mother trees (A33 and A39) to avoid inbreeding depression (Table 1). Reciprocal crosses between P236 and P336 were performed in May 2011, and the cones were harvested in September 2011 [19]. In May 2012, the seeds were sown in the nursery of The University of Tokyo Hokkaido Forest (UTHF), Furano, Japan (43°30′ N, 142°18′ E, 230 m asl). After the seedlings were grown for two successive growing seasons, 252 saplings were transferred to long pots (2320 cc) on 12 May 2015. The growing medium in the pots was a mixture of soft, solidified black soil (termed “kuroboku”), cocopeat (Top Co., Osaka, Japan), local volcanic ash soil, and bark compost in a 20:20:30:15 ratio (*v*/*v*/*v*/*v*).

The saplings were grown for another two consecutive growing seasons at the Forestry Research Institute, Hokkaido Research Organization (HRO), Bibai, Japan (43°17′ N, 141°51′ E, 40 m asl). Considering the environmental heterogeneity among microhabitats (e.g., sun exposure), periodic rotation of the pots was performed. Additionally, the saplings were irrigated every morning to prevent drought stress. The soil was watered with Osmocote^®^ Exact Standard 3-4M (N16-P9-K12) (ICL Specialty Fertilizers, Geldermalsen, The Netherlands) at the start of each growing season and with HYPONeX liquid (N6-P3360-K5) (HYPONeX Japan Co., Ltd., Osaka, Japan) during the latter part of the growing season, following the manufacturers’ protocols.

The saplings were subsequently transplanted to a common garden that was established at the UTHF (43°13′ N, 142°24′ E, 230 m asl) on 27 April 2017. The saplings were planted in a randomized order at 2.0 m intervals. No irrigation or fertilization was performed for the transplants in the common garden, only weed control.

### 2.2. Phenotyping

The phenotype data of 15 functional traits, including four growth and 11 eco-physiological traits, for the 252 saplings in the segregation population were recorded (Table 2). On 4 October 2016, the stem diameter (D16, mm) and height (H16, cm) of the saplings at the HRO were measured. After transplanting to the UTHF, the tree heights (H17, cm) were measured in the autumn of 2017, and images via photograph-based projection were captured from the top points of each tree. Then, the crown area (CR17, cm^2^) was estimated using LIA32 software (http://www.agr.nagoya-u.ac.jp/~shinkan/LIA32/) (Figure 1).

### 2.3. Characterization of Morphological Traits

One-year-old needles were randomly collected from the middle portion of each branch and scanned using a digital scanner (MSC20; King Jim, Tokyo, Japan). The mean length and width of the needles (Lw_ratio) were determined using ImageJ software (NIH, Bethesda, MD, USA; http://rsbweb.nih.gov/ij/). The leaf mass per area (LMA, g cm^−2^) was calculated by dividing the dry mass of 20 needles by the total needle area using ImageJ software. Following a previously used method [17], the stomatal density (SD, number mm^−2^) was estimated as the ratio of the number of stomata to the respective stomatal band area and the number of stomatal rows (SRN, number mm^−1^) was measured from the same images in a 0.88 mm × 0.66 mm area. Transverse sections of the current season shoots of the saplings were photographed using a digital camera (DP71; Olympus, Tokyo, Japan) mounted on a light microscope (BX50; Olympus). The images were used to measure the ratio of bark width to xylem width (Bark_xy) and the ratio of normal wood width to reaction wood width (Norm_reac) using ImageJ software.

### 2.4. Evaluation of Bud Phenology

As a trait associated with plant phenology, the timing of bud flush (Bud_fl) was also recorded. On 6–29 May 2016, the status of the terminal bud in each pot sapling was monitored every two days. Following a previous study [19], Bud_fl was defined as the time point at which the initial emergence of needles from the terminal bud was observed. Hence, the number of days after 1 May was recorded as the Bud_fl.

### 2.5. Evaluation of Freezing Tolerance

To measure freezing tolerance, a freezing test was conducted on November 4, following the method of a previous study on Sakhalin fir [25]. Before the test, three healthy current-year needles were collected in the morning, placed in a thin plastic bag (0.03 mm), and incubated for >3 h in a dark chamber maintained at 4 °C for dark acclimation. Each test was conducted by exposing the needles to the target temperature (−20 °C) for 5 h, and then decreasing temperature at a rate of 12 °C h^−1^ using a programmed freezer (SC-DF25K; Nippon Freezer Co., Tokyo, Japan). We measured the freezing damage to the needles after incubation for 2 days at 4 °C, followed by an increase in temperature at a rate of 8 °C h^−1^ using a pulse-amplitude modulated (PAM) chlorophyll fluorometer (Mini-PAM; Walz, Effeltrich, Germany). The maximum photochemical quantum yield of photosystem II (PSII; Fv/Fm) is negatively correlated with the extent of freezing damage, and thus, represents a reliable parameter for the assessment of freezing tolerance during autumn [25,26]. The Fv/Fm values used as an index of freezing tolerance in this study were termed “Freez_tol”.

### 2.6. Measurement of Chlorophyll Fluorescence

The saplings at the HRO were placed in a darkroom for 3 days. Chlorophyll fluorescence was then measured with a chlorophyll fluorometer (Junior-PAM; Walz, Effeltrich, Germany) using a detached leaf from the major axis of a current-season branch. The measurements were performed in a ventilated room, with 40 Pa CO_2_ and 21 kPa O_2_ at 5 °C. Saturation pulses from blue light-emitting diodes (LEDs; >6000 μmol photons m^−2^ s^−1^, 800 ms duration) were applied to determine the maximum chlorophyll fluorescence, with closed PSII centers in the dark (Fm) and under actinic light (Fm′). The maximum photochemical quantum yield (Fv/Fm) and effective quantum yield (Φ_II_) of PSII were calculated as (Fm–F0)/Fm and (Fm′–Fs′)/Fm′, respectively [27], where F0 is the minimal chlorophyll fluorescence yield in the dark and Fs is the steady-state chlorophyll fluorescence level under actinic light from blue LEDs, with a wavelength peak at 445 nm. Non-photochemical quenching (NPQ) was calculated as (Fm–Fm′)/Fm′ [28]. Additionally, the PSII quantum yields Φ_NPQ_ and Φ_NO_ [29] that represent the regulated and non-regulated energy dissipation at the PSII centers, respectively, and add up to the unity of the photochemical quantum yield (i.e., Φ_II_ + Φ_NPQ_ + Φ_NO_ = 1), were also calculated. The values of Φ_NPQ_ and Φ_NO_ were calculated as Fs′/Fm–Fs′/Fm and Fs/Fm, respectively [30,31]. The Fv/Fm in dark-adapted leaves was measured overnight, and then the other parameters were measured under actinic light of 1500 μmol photons m^−2^ s^−1^ for 2 min.

#### Determination of Relationships between Phenotypic Traits

The relationships between the functional traits were analyzed using R version 3.6.1 [32]. The Pearson’s correlation coefficients and *p* values for correlations between functional traits were calculated. To simplify and visualize the strong correlations, a network plot was generated using the R package ‘corrr’ version 0.4.3 [33]. The proximity of the points was determined using multidimensional clustering, and the minimum coefficient value (in absolute terms) was set to 0.1.

### 2.7. Construction of Double-Digest Restriction Site-Associated DNA Sequencing (ddRAD-seq) Library

A total of 252 individuals were included in ddRAD-seq experiment [34]. Compared with the original restriction site-associated DNA sequencing (RAD-seq) [35], ddRAD-seq effectively reduced the complexity of studying the large genome of conifers [36]. The total genomic DNA (250 ng) from each sample was digested with *Sph*I and *Pst*I, ligated with Y-shaped adaptors, and amplified by PCR using KAPA HiFi polymerase (KAPA Biosystems, Woburn, MA, USA). After PCR amplification with adapter-specific primer pairs (Access Array Barcode Library for Illumina Sequencers; Fluidigm, San Francisco, CA, USA), an equal amount of DNA from each sample was mixed and size-selected using BluePippin agarose gel cassettes (Sage Science, Beverly, MA, USA). The library fragments (~450 bp) were retrieved, and the quality of the library was checked using an Agilent 2100 Bioanalyzer with a high-sensitivity DNA chip (Agilent Technologies, Waldbronn, Germany). The library was sequenced using the Illumina^®^ HiSeq X platform (Illumina, San Diego, CA, USA) to generate 150 bp long paired-end reads (see details in Appendix A). The raw ddRAD-seq data were deposited in the DNA Data Bank of Japan (DDBJ) under accession number DRA012397.

### 2.8. Genotyping

Quality control, adapter trimming, and quality filtering of the ddRAD-seq raw data were performed using fastp program version 0.20.0 [37]. A sliding window of 15 bases was used to filter reads with low-quality scores (Phred quality values < 10). The high-quality ddRAD-seq data were processed using the ustacks, cstacks, sstacks, tsv2bam, and gstacks programs of Stacks software version 2.5 [38], following the Stacks manual for a *de novo* genetic mapping cross (http://catchenlab.life.illinois.edu/stacks/manual/, accessed on 21 July 2021). This method builds catalogs from the parents of a mapping population only, which enables the exclusion of potential errors derived from the mapping progeny. The number of mismatches allowed between stacks within and between individuals was set to four (i.e., M and n options in ustacks and cstacks, respectively). The populations program in Stacks pipeline was used to retain the loci that were observed in >80% of individuals in the mapping progeny. The retained loci were exported in JoinMap format via a cross pollination (CP) mode using the same program to construct the linkage maps.

### 2.9. Linkage Map Construction

Two types of linkage maps (P336 and P236) were constructed based on the segregation patterns of the genetic markers, following the method of the previous study [19]. The P336 and P236 maps were comprised of the aa × ab and ab × aa type markers, respectively. Markers that contained a large amount of missing data (>20%) and showed significant deviation (chi-square test; *p* < 0.001) from the expected Mendelian segregation ratio were excluded. Linkage analysis was conducted for the populations of cross pollinators (CP) with the regression mapping method using JoinMap v5.0 software [39]. Default settings were used for determining the recombination frequency.

### 2.10. QTL Analysis

QTL analysis was conducted using linear models with a regularized horseshoe prior to distribution, which can detect and assess relevant variables from a large number of irrelevant variables [40]. First, to reduce collinearity and redundancy among variables, one marker from a given pair of markers that were in close proximity (<1 cM) and showed high correlation (*r* > 0.7) with other markers was removed prior to the analysis. Second, four chains were used to estimate the posterior distribution, and a total of 5000 iterations after the first 1000 iterations were discarded as burn-in. The convergence of the chains was confirmed using the Gelman–Rubin statistic ($\hat{R}$ < 1.1). Third, the posterior distribution was estimated using the R package ‘rstanarm’ [41]. The global shrinkage parameter of the regularized horseshoe prior was set using the equation proposed by a previous study [40]:$$\frac{p0}{D-p0}\frac{1}{\sqrt{n}}$$
where *p*0 is the expected number of relevant variables, *D* is the number of variables, and *n* is the number of observations. For all traits, the value of *p*0 was set to five because the QTLs of various tree species were known to exhibit small effects [42]. The sensitivity of the *p*0 value for QTL detection was confirmed by arranging *p*0 from 1 to 9 (Appendix A). The default values of the regularized horseshoe prior in the R package ‘rstanarm’ were used for the remaining parameters.

Fourth, given that the variables in the aforementioned model were highly correlated, the estimated marginals of the posterior distribution of the marker effects may be biased because of collinearity. To address this problem, a model selection method, projection-predictive variable selection [43], was applied to the model. The model selection method uses a model that has the best predictive power (a reference model) to find a simpler model that has similar predictions as the reference model (predictive projection). The linear model described above was used as the reference model. Model selection was performed when the variables in the reference model contained significantly more information than the null model, which was confirmed by leave-one-out cross-validation [44]. When the marginal posterior distribution of the marker effects in the selected model did not contain zero in the 95% and 80% credible intervals, the markers were defined as significant and suggestive of QTLs, respectively. The contribution of each QTL was estimated by the coefficient of determination (R^2^) of the simple regression; and the contribution was defined as the percentage variance explained (PVE) for a QTL. The QTL analysis and model selection were both performed using R version 3.6.1 [32]. The leave-one-out cross-validation procedure was performed using the R package ‘loo’ [45], while the model selection method was conducted using the R package ‘projpred’ [43].

### 2.11. Candidate Gene Prediction

Genes in the target QTL regions were predicted in the Sakhalin fir transcriptome database TodoFirGene [20] using the basic local alignment search tool (BLAST) nucleotide algorithm. The transcripts of the candidate genes were obtained from TodoFirGene and subsequently annotated using the Gene Ontology (GO) and Kyoto Encyclopedia of Genes and Genomes (KEGG; https://www.kegg.jp/kegg/kegg2.html, accessed on 16 July 2021) databases.

## 3. Results

### 3.1. Phenotypic Traits

Based on the results of previous studies, these traits are potentially associated with altitudinal adaptation (Table 1). The mean values, standard deviations, and coefficients of variation for the 15 functional traits are presented in Table 1. The distribution of each phenotype was observed to have a modal shape (Appendix A). The network analysis revealed the relationships among the functional traits investigated, in which two clusters were identified for the growth and photosynthetic traits (Figure 2). Furthermore, growth-related traits were discovered to be positively correlated with Bud_fl, whereas Lw_ratio was negatively correlated with Bark_xy ratio. Pairwise correlations between functional traits using Pearson’s correlation coefficients showed significant negative correlations between certain photosynthetic traits, such as Φ_II_ vs. NPQ (*r* = −0.63, *p* < 0.001) and NPQ vs. Φ_NO_ (*r* = −0.89, *p* < 0.001) (Appendix A). However, a positive correlation was observed between Φ_II_ and Φ_NO_ (*r* = 0.24), whereas strong and highly positive correlations were observed between growth traits (*r* > 0.5).

### 3.2. Linkage Map Construction and QTL Detection

A total of 486 and 516 markers were mapped to 12 linkage groups (LG) in the P336 and P236 maps, respectively. For the P336 and P236 maps, the total lengths were 1986.2 cM and 1932.8 cM, respectively, whereas the average distances between adjacent markers were 4.1 cM and 3.7 cM, respectively. The maximum marker gaps for P336 map and P236 map were 29.8 cM and 27.3 cM, respectively (Table 3).

Three significant and 11 suggestive QTLs were detected in the P336 map, whereas five significant and 10 suggestive QTLs were observed in the P236 map (Table 4; Figure 3). Significant QTLs for Φ_NO_ and CR17 and for Φ_NO_, H16, H17, and CR17 were detected in the P336 and P236 maps, respectively. In addition, suggestive QTLs for NPQ, Bud_fl, Freez_tol, Lw_ratio, H16, H17, and CR17 and for NPQ, Bud_fl, Lw_ratio, D16, H17, and CR17 were detected in the P336 and P236 maps, respectively. A significant QTL (Locus #2055) was detected for NPQ and Φ_NO_ in both linkage maps. In the P236 map, Locus #12865 was discovered as a significant QTL for CR17 and a suggestive QTL for D16, H16, and H17 (Figure 3). In the P336 map, suggestive QTLs for Bud_fl and Freez_tol were co-located in LG5 within a 10 cM distance (Figure 3).

### 3.3. Candidate Gene Prediction

Out of 16 unique QTLs, eight were found in the transcripts of TodoFirGene (Appendix A). Among the eight QTLs, locus #1970, which is associated with CR17, was annotated as a type-A *Arabidopsis* response regulator (type-A ARR) of phosphorelay signal transduction system and a type-A ARR of cytokinin signaling in the GO and KEGG databases, respectively. The sequence of locus #1970 was shown in Appendix A.

## 4. Discussion

### 4.1. Segregated Population and Linkage Maps

Long generation times and the outbred mating system of conifers were used to construct a suitable mapping population. Because several pedigrees are available for important traits, including height growth and wood properties, in a tree breeding program, second-generation populations were used for QTL mapping [46,47,48,49]. However, F_1_ pedigrees have been used for QTL mapping of eco-physiological studies in the evolutionary biology of conifers [50,51]. In the present study, we used segregated populations based on control crosses with four different grandparent trees with distinct genetic backgrounds (Table 1) [19]. Furthermore, altitudinal adaptation was clearly demonstrated by a reciprocal transplant experiment [18]. Our studies are limited by the number of pedigrees, locations and sample replications. Nevertheless, the segregated populations were constructed by a control cross between F_1_ hybrids with two contrasting ecotypes (high-altitude and low-altitude grandparents), which might be useful to clarify the genetic basis of altitudinal adaptation in conifers as described in a previous paper [19].

The linkage maps were saturated, with a 10 cM interval between nearest markers. The average distance between nearest markers detected in this study (approximately 4 cM; Table 3) was also comparable with that in previous studies [46,48,50,52]. The maximum marker gap distance in this study (approximately 20 cM) was also similar to that in a previous study [51]. Recently, high-density linkage maps with an average distance between nearest markers of less than 1 cM have been constructed with several thousand SNP markers [47,53]. In addition, gene-based markers have been applied to obtain functional information [50]. The reconstruction of the linkage map using gene-based markers might improve the accuracy of QTL mapping in future studies.

### 4.2. Growth Traits

The segregation populations reflected the broad growth variations along an altitudinal gradient. The coefficient of variation (CV) for the heights of 4-year-old Sakhalin fir seedlings was ~0.20, which is similar to the CV value (0.19) reported for 4-year-old seedlings in the provenance trials [16]. The growth traits analyzed in the present study were mostly controlled by several QTLs with small or moderate effects (PVE: 2.6–9.3%) (Figure 3; Table 4). This result corroborates that of a previous study on Sakhalin fir, in which several QTLs with small or moderate effects (PVE: 3.8–7.4%) were detected for H17 and D16 at the early stage. QTLs with small or moderate effects on growth traits have also been detected in other conifers [46,47,48,49].

The crown area and width seem to be a good indicator for growth. In the present study, CR17 was measured using the digital images for growth traits (Figure 1). The number of QTLs for CR17 was the highest among the traits examined in this study; five exhibited moderate effects (PVE > 5%; Table 2; Figure 3). The QTLs for crown width detected in this study have also been reported in other conifers [47]. Notably, Locus #1970 of the QTLs for CR17 was annotated as a type-A *Arabidopsis* Response Regulator (type-A ARR; Appendix A), which functions as a key node in the integration of ethylene and cytokinin signaling to regulate plant responses and shoot elongation in *Arabidopsis thaliana* [54]. Type-A response regulators (Type-A RRs) play an important role in cytokinin-induced adventitious shoot formation in Pinaceae (*Pinus pinea*) [55]. Furthermore, it is known that crown size is determined by the expansion and division of cells in the apical and cambial meristems. Yang et al. [47] previously detected QTLs for growth traits, including crown width, in the interspecific hybrid clones of *Taxodium* species, which is widely grown in southern China for economic and ecological purposes, and reported that the three markers were annotated as a leucine-rich repeat receptor-like kinase (*LRR-RLK*) gene, which is specifically expressed in the vascular tissues and is vital for stem growth.

### 4.3. Eco-Physiological Traits

Among the morphological traits, three QTLs were detected for needle length/width ratio (Table 4; Figure 3). The provenance trial of Sakhalin fir demonstrated that the length/width ratio was negatively correlated with the altitude of seed source [17]. However, no studies related to the QTLs for needle length/width ratio were available for other conifers. For example, needle length was only investigated in *Pinus elliottii* var. *elliottii* × *P. caribaea* var. *hondurensis* hybrids [48]. In *Populus* and European beech, a major QTL for leaf length/width ratio was identified using linkage mapping [56,57]. No QTLs were detected for the other morphological traits (i.e., Norm_reac, LMA, SD, and SRN) examined in the present study. However, altitudinal clines have been detected in mature stands based on a provenance trial [17], suggesting that these morphological trends may have emerged at an advanced life-cycle stage.

Additionally, QTLs were detected for Bud_fl and Freez_tol (Table 4; Figure 3). A QTL for Bud_fl was also detected in a previous study of Sakhalin fir [17]. Bud phenology traits generally exhibit high heritability and are often controlled by several QTLs. A positive correlation between Bud_fl and growth traits was observed, indicating that the saplings with later Bud_fl tended to be larger in growth, which supports the data of a previous study on Sakhalin fir [19]. The Freez_tol was also found to be positively correlated with growth traits (Figure 2; Appendix A). The timing of acquiring Freez_tol was associated with the altitudinal adaptation of Sakhalin fir [25,58]. The locations of the QTLs for Bud_flush and Freez_tol were relatively close in the P336 map-LG5, with a distance of ~10 cM (Figure 3). QTL clusters for height, density, and wood properties were also previously reported in Douglas fir [52]. These QTL clusters potentially represent pleiotropic effects or may be evidence for clusters of linked genes [59]. Because generating large amounts of meaningful data related to photosynthesis is easy, the data on chlorophyll fluorescence, including Φ_II_, NPQ, and Φ_NO_, can be directly applied to eco-physiological studies [60]. Φ_II_ is an indicator of photosynthetic activity [29]. Because NPQ is associated with functional traits required to avoid photoinhibition [61] and Φ_NO_ represents the non-regulated energy dissipation at the PSII centers, these parameters might be important for trees living in high-altitude zones. Although we did not detect any QTLs for Φ_II_, QTLs for NPQ and Φ_NO_ were readily observed (Table 4; Figure 3). In general, the inhibition of photosynthesis by low temperature results in a considerable excess of energy, leading to photodamage. Specifically, selection to avoid photoinhibition should be stronger than that in low-altitude zones. Therefore, energy dissipation of heat and non-regulated energy might be essential for adaptation in high-altitudinal zones.

## 5. Conclusions

To elucidate the genetic basis of altitudinal adaptation in Sakhalin fir, a QTL analysis with linkage maps was conducted for 15 functional traits potentially associated with altitudinal adaptation. Similar to a previous study, QTLs with moderate PVE were detected for growth traits; ten were linked to CR17. Notably, one QTL for CR17 might function as a type-A RR, which plays an important role in cytokinin-induced shoot elongation. The QTLs for morphological traits (needle length/width ratio) and phenology (Bud_fl and Freez_tol) were also identified. Furthermore, our results suggest that two significant QTLs may be genetically controlling Φ_NO_ in Sakhalin fir. Reproducibility of the QTLs detected in the present study should be examined in the future because we conducted the analysis with a single pedigree, in a single location, and there was no replication of segregated populations. Nevertheless, we believe that these findings might be an important initial step for detecting candidate genes for altitudinal adaptation in conifers because altitudinal adaptation has been demonstrated by a reciprocal transplant experiment [18], and a segregation population was produced by a control cross between F_1_ hybrids with grandparents that were high-altitude and low-altitude genotypes [19].

## Figures and Tables

**Figure 1 genes-12-01110-f001:**
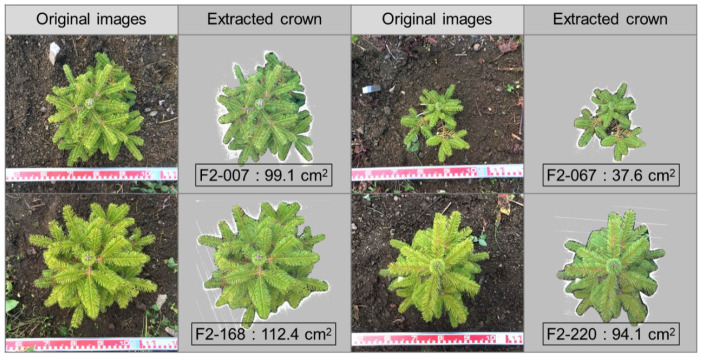
The original images taken via photograph-based projection (**left**) and the images of the individual plants with the calculated crown area based on the scale (**right**).

**Figure 2 genes-12-01110-f002:**
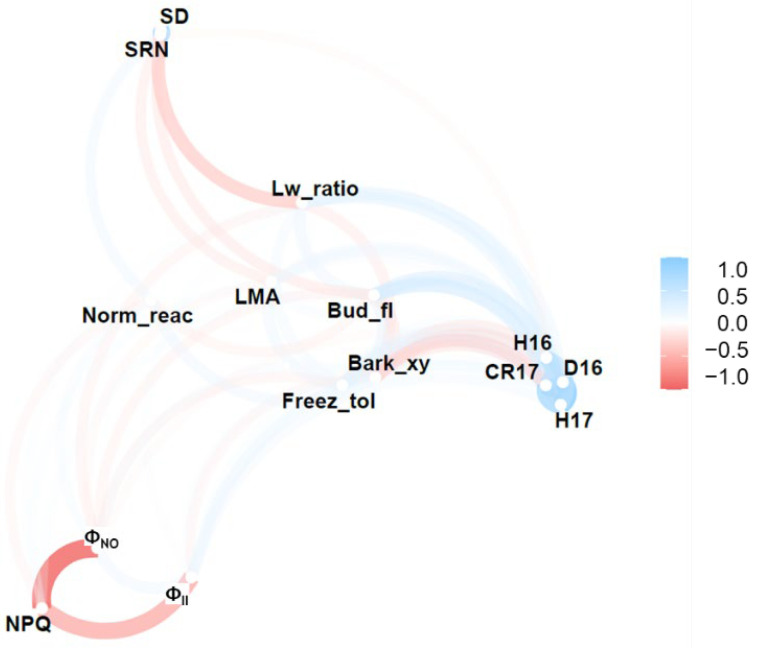
The relationships between functional traits based on network analysis. D16: Stem diameter in 2016, H16: Height in 2016, H17: Height in 2017, CR17: Crown area in 2017, Bud_fl: Bud flush in 2016, Freez_tol: Freezing tolerance in Nov 2016, Lw_ratio: Needle length/width ratio, LMA: Leaf mass per area, Bark_xy: Bark-xylem length ratio, Norm_reac: Normal/reaction wood ratio, SD: Stoma density, SRN: Number of stoma row, Φ_II_: Effective quantum yield of PSII, NPQ: Non-photochemical quenching, Φ_NO_: Non-regulated energy dissipation at PSII centers.

**Figure 3 genes-12-01110-f003:**
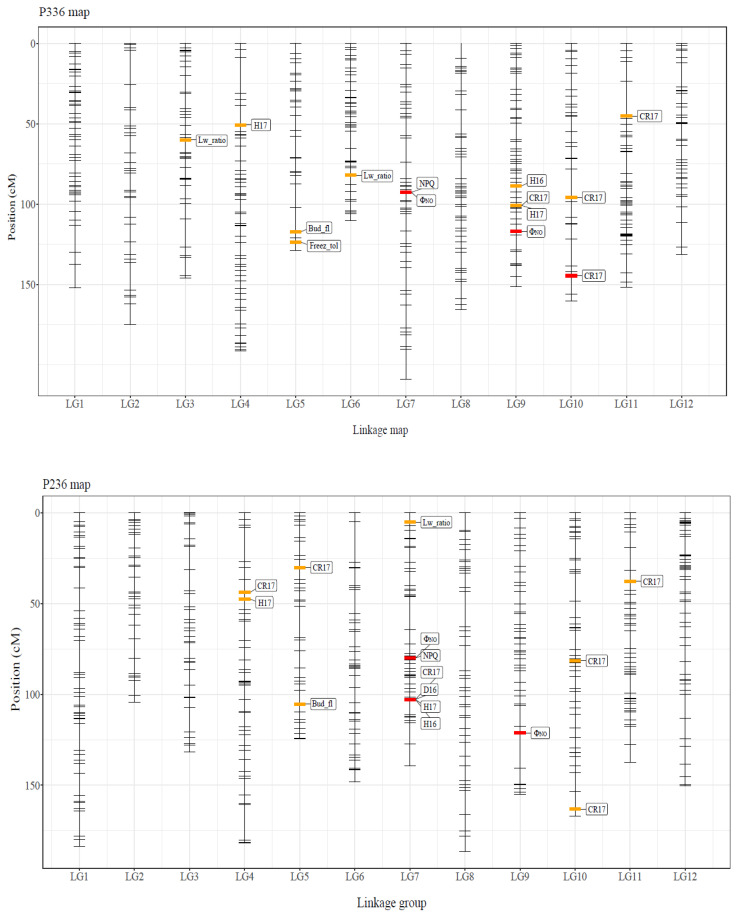
QTL mapping for the 15 functional traits. P336 and P236 maps were demonstrated in the upper and lower panels, respectively.

**Table 1 genes-12-01110-t001:** The Sakhalin fir segregated population used for QTL analysis. Grandmother parent Clones (A33 and A39) are mother trees inhabiting high-altitude zones (1200 m). Clone 1-1 and Clone 1-2 are low-altitude genotypes derived from a pollen mixture collected from three genotypes living at low-altitudes (530 m).

Pedigree	Female × Male Parent	Female × Male Parent
Grandparents	Clone A33 × Clone C1-1 ^†^	Clone A39 × Clone C1-2 ^†^
Parents	Clone P336 × Clone P236, Clone P236 × Clone P336	
Parent size	P336 (height: 15.5 m, d.b.h.: 21.1 cm)	
	P236 (height: 16.4 m, d.b.h.: 23.4 cm)	
Crossing	May 2011	
Seed collection	September 2011	
Progeny	252	

^†^ Clone 1-1 and Clone 1-2 are different genotypes confirmed by paternity analysis with nSSR and cpSSR markers. Height and diameter at breast height (d.b.h.) were measured in October 2018.

**Table 2 genes-12-01110-t002:** The 15 functional traits investigated in this study. The correlation between functional traits and altitude was demonstrated in previous studies of Sakhalin fir. The mean values, standard deviations (SD), and coefficients of variation (CV, %) are presented. ^†^ Positive: LMA is generally highly correlated with leaf thickness.

Category	Trait Abberviations	Trait Explanation	Correlation with Altitude	Reference	Mean (SD)	CV
Growth	D16	Stem diameter in 2016	Negative	[16]	35.01 (8.595)	0.246
	H16	Height in 2016	Negative	[16]	11.39 (2.298)	0.202
	H17	Height in 2017	Negative	[16]	45.15 (10.94)	0.242
	CR17	Crown area in 2017	Unknown	-	106.3 (39.43)	0.371
Phenology	Bud_fl	Bud flush in 2016	Unknown	-	20.05 (2.538)	0.127
	Freez_tol	Freezing tolerance in Nov 2016	Positive	[25]	0.387 (0.209)	0.541
Morphology	Lw_ratio	Needle length/width ratio	Negative	[17]	19.21 (3.277)	0.171
	LMA	Leaf mass per area	Positive^†^	[17] (Thickness)	0.112 (0.024)	0.211
	Bark_xy	Bark-xylem length ratio	Positive	[17]	0.452 (0.042)	0.093
	Norm_reac	Normal/reaction wood ratio	Negative	[17]	0.604 (0.113)	0.188
	SD	Stoma density	Unknown	-	289.8 (54.34)	0.188
	SRN	Number of stoma row	Positive	[17]	11.80 (1.583)	0.134
Photosynthesis	Φ_II_	Effective quantum yield of PSII	Unknown	-	0.308 (0.043)	0.138
	NPQ	Non-photochemical quenching	Unknown	-	2.483 (0.513)	0.207
	Φ_NO_	Non-regulated energy dissipation at PSII centers	Unknown	-	0.202 (0.024)	0.117

**Table 3 genes-12-01110-t003:** The length of and the number of markers in each linkage group.

Map	Linkage Group	Marker	Length (cM)	Average Distance between Markers (cM)	Gap (Max.)
P336	1	42	154.4	3.7	17.3
	2	30	174.9	5.8	21.8
	3	31	145.6	4.7	17.6
	4	50	192.9	3.9	22.2
	5	28	128.6	4.6	15.8
	6	40	147.5	3.7	29.8
	7	44	208.1	4.7	18.0
	8	45	162.5	3.6	14.1
	9	51	151.3	3.0	10.1
	10	36	192.6	5.4	26.3
	11	45	151.8	3.4	20.9
	12	44	176.0	4.0	27.8
	Total	486	1986.2	4.1	29.8
P236	1	51	185.0	3.6	18.7
	2	45	189.5	4.2	27.3
	3	35	131.7	3.8	13.7
	4	44	180.7	4.1	19.9
	5	31	126.6	4.1	17.4
	6	39	141.3	3.6	22.2
	7	54	180.1	3.3	18.5
	8	43	185.8	4.3	18.5
	9	43	155.2	3.6	19.0
	10	44	166.1	3.8	18.3
	11	43	136.4	3.2	12.5
	12	44	154.7	3.5	13.7
	Total	516	1932.8	3.7	27.3

**Table 4 genes-12-01110-t004:** The QTLs detected for each functional trait. The marker position (Pos.) and percentage variance explained (PVE) of each QTL are shown. The level of significance is presented in the Sig. column, with * 80% and ** 95% confidence intervals.

Category	Trait	Locus	Map-LG	Pos (cM)	Sig.	PVE (%)
Growth	D16	#12865	P236-LG7	102.8	*	6.70
	H16	#10164	P336-LG9	88.7	*	4.27
	H16	#12865	P236-LG7	102.8	**	9.17
	H17	#10758	P336-LG4	50.7	*	4.16
	H17	#10541	P336-LG9	100.9	*	4.57
	H17	#10758	P236-LG4	47.5	*	4.16
	H17	#12865	P236-LG7	102.8	**	7.81
	CR17	#6809	P336-LG10	95.9	*	2.62
	CR17	#10829	P336-LG10	144.6	**	5.66
	CR17	#1970	P336-LG11	45.2	*	5.45
	CR17	#10541	P336-LG9	100.9	*	4.28
	CR17	#6809	P236-LG10	81.5	*	2.62
Phenology	Bud_fl	#6899	P336-LG5	117.2	*	6.07
	Bud_fl	#6899	P236-LG5	105.4	*	6.07
	Freez_tol	#25432	P336-LG5	123.8	*	7.40
Morphology	Lw_ratio	#24342	P336-LG3	60.0	*	5.96
	Lw_ratio	#7510	P336-LG6	82.0	*	2.97
	Lw_ratio	#30964	P236-LG7	5.2	*	5.28
Photosynthesis	NPQ	#2055	P336-LG7	92.6	*	6.12
	NPQ	#2055	P236-LG7	80.1	*	6.12
	Φ_NO_	#2055	P336-LG7	92.6	**	9.32
	Φ_NO_	#27288	P336-LG9	117.1	**	7.24
	Φ_NO_	#2055	P236-LG7	80.1	**	9.32
	Φ_NO_	#27288	P236-LG9	121.3	**	7.24

## Data Availability

The raw ddRAD-seq data generated in this study were deposited in the DNA Data Bank of Japan (DDBJ) under accession number DRA012397.

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
