# Peer review of "Genetic Dissection of Growth and Eco-Physiological Traits Associated with Altitudinal Adaptation in Sakhalin Fir (Abies sachalinensis) Based on QTL Mapping"

_genes, 2021, doi:10.3390/genes12081110_

Round 1

Reviewer 1 Report

 Dear authors.

This paper is a study of QTL analysis to identify traits and their genes for local adaptation. Reviewing this paper, I have little to comment on the presentation of the ''introduction,'' ''materials and methods,'' and ''Results. However, I do have some comments on the trait assessment for QTL analysis of conifers.

Conifer species are rarely approached with a genetic background for detailed evaluation of traits and their underlying functional relationships because of their material and age. Therefore, it is necessary to be cautious about the interpretation of the results of QTL analysis and related results.

(1) The authors show that in L343-349, one of the genes identified was the ARR3 gene and their thoughts on it. However, from the literature cited, the function of the ARR3 gene shown is for Arabidopsis, and there appears to be no evidence for it in Sakhalin fir. The gene identified is an important gene, but the statement that this gene appears to be acting definitively on this trait should be revised.

 (2) Similarly, in L375-383, the reason why QTLs related to photosynthesis were not identified is described. However, there is a need to first accumulate physiological studies to see if the evaluation of these traits was appropriate. Therefore, I believe that the description in this section should also be revised. It is difficult to understand that the data were obtained based on an appropriate evaluation method simply because it is easier to measure photosynthesis compared to the process of selecting Sakhalin fir as the material for local adaptation.

Author Response

Dear Reviewer 1

Thank you for your helpful suggestions. We significantly revised manuscript as below.

(1) The authors show that in L343-349, one of the genes identified was the ARR3 gene and their thoughts on it. However, from the literature cited, the function of the ARR3 gene shown is for Arabidopsis, and there appears to be no evidence for it in Sakhalin fir. The gene identified is an important gene, but the statement that this gene appears to be acting definitively on this trait should be revised.

Response: Thank you for helpful suggestions. Yes, a Type-A ARR (Arabidopsis Response Receptor) gene was detected in Arabidopsis. We understand that our phrasing could be misleading for readers. However, transcript ID “AbisacEGm029060t1” used in this study was derived from RNA-seq data of Sakhalin fir. We used sequences including SNP markers for QTLs derived from ddRAD-seq to predict the annotation of candidate genes based on the transcriptome database of Sakhalin fir, TodoFirGene (http://plantomics.mind.meiji.ac.jp/todomatsu/). To enable readers to confirm the process, we added new supplemental material (Appendix A2) for the sequence of locus#1970. Recently, type A response receptor (type-A RR) genes were detected in Pinaceae. Therefore, we revised this text according to the comments (L311-315). 

 (2) Similarly, in L375-383, the reason why QTLs related to photosynthesis were not identified is described. However, there is a need to first accumulate physiological studies to see if the evaluation of these traits was appropriate. Therefore, I believe that the description in this section should also be revised. It is difficult to understand that the data were obtained based on an appropriate evaluation method simply because it is easier to measure photosynthesis compared to the process of selecting Sakhalin fir as the material for local adaptation.

Response: Thank you for helpful suggestions. Not only Φâ…¡ but also NPQ and ΦNO are important parameters for photosynthesis. Because NPQ is associated with functional traits required to avoid photoinhibition and ΦNO represents the non-regulated energy dissipation at the PSII centers, these parameters might be important for trees living at high-altitude zones. In general, the inhibition of photosynthesis by low temperature results in a considerable excess of energy leading to photodamage. Namely, selection to avoid photoinhibition should be stronger than that in low-altitude zones. We have revised the manuscript, explaining that selection in high-altitude zones might be stronger than that in low-altitude ones. Accordingly, QTL of NPQ might be detected in the segregated population (L338-346).  

Reviewer 2 Report

Review on: “Genetic dissection of growth and ecophysiological traits associated with altitudinal adaptation in Sakhalin fir (Abies sachalinensis) based on QTL mapping” by Susumu Goto et al. is devoted to studies of multiple traits based on two mapping populations using NGS-based markers. The genetic maps were reasonably saturated with molecular markers. However, some gaps were present. The authors have evaluated relatively large mapping populations during several year-long experiments, and the traits were investigated in a single location during one season. QTL mapping allowed identifying QTLs for most of the investigated traits; however, only in some cases did the QTLs’ effects substantial.
Comments:
1. As not everyone works in the genetic mapping of trees, it would be of value to include the scheme of how the P1 and P2 were derived. 
2. What do you mean by “populations program”? A reference is required and the program title.
3. Have you checked for the normality of the traits? What was the segregation ratio of the traits? Any guess on the number of QTLs you might have expected? Please, present the issue in Results and discuss it.
4. Having genetic maps, it is recommended to calculate linkage disequilibrium to ensure that marker saturation is sufficient for QTL analysis. Please, calculate LG.
5. Additional information on genetic maps (Gaps, number of markers per cM, marker density, number of redundant markers etc.) should be added (in Table 3). The results should be compared to other genetic maps of the species.
6. I would suggest considering performing association mapping on the whole set of markers to verify QTLs position and identify additional markers of the traits. Furthermore, I suggest supporting information on traits’ heritability (narrow sense).
7. As your QTL mapping (due to how the mapping populations were constructed – limited initial variation to parental forms) might be subjected to parental effect, please discuss the issue in relation to low and moderate QTLs.
8. As your QTL experiment was conducted in a single location with no repetitions, please discuss the issue in the context of the QTLs and their significance (how can you rely on your results, how the others cope with the same problem, what is the reproducibility of such results – please, try indirectly support the notion that your results reflect real cases rather than some fluctuations in the case of QTLs with low and maybe moderate effects).
Conclusions
In general, the study is performed correctly, but additional analysis is needed. The disadvantage is that some QTLs are instead a week and thus, might not be reliable. At least indirect confirmation of the importance of such QTLs is requested in the discussion. For other comments, please refer to the text above. 

Author Response

Dear Reviewer 2

Thank you for your helpful suggestions. We significantly revised the manuscript according to the comments.

Comments:
1. As not everyone works in the genetic mapping of trees, it would be of value to include the scheme of how the P1 and P2 were derived. 

Response: Thank you for your helpful comments. We added a new table (Table 1 in the current version) for the segregated populations including information on parents and grandparents. We changed the terms “P1 and P2 maps” to “P336 and P236 maps,” referring to the previous study (Goto et al. 2017).

  1. What do you mean by “populations program”? A reference is required and the program title.

Response: The populations program is a setting in the Stacks pipeline mentioned just above (L181–183). We revised that sentence for clarity (L186–187).

  1. Have you checked for the normality of the traits? What was the segregation ratio of the traits? Any guess on the number of QTLs you might have expected? Please, present the issue in Results and discuss it.

Response: We conducted QTL analysis using the linkage map of the bi-parental progeny. In this study, we assumed that phenotypic traits are not polygenic but have an oligogenic distribution. Therefore, we need not check the normality of the phenotypic traits in this study. Furthermore, because all the phenotypic values are continuous, the segregation ratio cannot be calculated by using them. In addition, we cannot estimate the number of QTLs due to lack of whole genome sequence information because conifers are not model organisms like Arabidopsis thaliana.

  1. Having genetic maps, it is recommended to calculate linkage disequilibrium to ensure that marker saturation is sufficient for QTL analysis. Please, calculate LG.

Response: Based on the recombination rate, we conducted QTL analysis using the biparental segregation population. To conduct a genome-wide association study, the linkage disequilibrium would be important. However, for QTL analysis with the linkage map in this study, we believe that linkage disequilibrium is not essential.

  1. Additional information on genetic maps (Gaps, number of markers per cM, marker density, number of redundant markers etc.) should be added (in Table 3). The results should be compared to other genetic maps of the species.

Response: According to your comments, we added additional information on the genetic maps to Table 3. We also included the results and compared them with other genetic maps of conifers in the Discussion (L292–298).

  1. I would suggest considering performing association mapping on the whole set of markers to verify QTLs position and identify additional markers of the traits. Furthermore, I suggest supporting information on traits’ heritability (narrow sense).

Response: In this study, we used a bi-parental progeny. We believe that association mapping is not suitable for this population because association mapping should be applied to natural populations or populations based on several pedigrees. In addition, narrow sense heritability of each trait cannot be calculated due to lack of replication of locations (environmental variation).

  1. As your QTL mapping (due to how the mapping populations were constructed – limited initial variation to parental forms) might be subjected to parental effect, please discuss the issue in relation to low and moderate QTLs.

Response: We agree with this comment. We can only use polymorphic data for the heterozygous loci of each parent. Therefore, the detection power of QTL mapping is dependent on parental polymorphisms. We added the limitations of this study to the Discussion (L288-291).

  1. As your QTL experiment was conducted in a single location with no repetitions, please discuss the issue in the context of the QTLs and their significance (how can you rely on your results, how the others cope with the same problem, what is the reproducibility of such results – please, try indirectly support the notion that your results reflect real cases rather than some fluctuations in the case of QTLs with low and maybe moderate effects).

Response: We agree with this comment. Actually, we planned to establish two locations using ca. 500 saplings for this population. However, the number of saplings finally obtained was only 252. To obtain the detection power per location, we established only one location for the segregated population. We have added discussion on the limitations of this study concerning the number of locations and replications (L353-359).

Conclusions

In general, the study is performed correctly, but additional analysis is needed. The disadvantage is that some QTLs are instead a week and thus, might not be reliable. At least indirect confirmation of the importance of such QTLs is requested in the discussion. For other comments, please refer to the text above. 

Response: Thank you for helpful suggestions. The PVE value of each QTL would be one indicator of reliability. As we have already demonstrated, the PVE of QTLs was concordant with the results of previous studies (L304–307, L355-359). Furthermore, as we described above, we understand the limitation of this study using one location with no replication. However, our bi-parental progeny was created by control crossing of F1 hybrids with grandparents of high-altitude and low-altitude genotypes. This is the strength of this study (Table 1 in the current version). Therefore, we stressed the importance of our segregated population (L289–291). The reliability of QTLs should be confirmed by future study. Unfortunately, rooted cutting with 10-years-old trees is quite difficult with Abies sachalinensis. We have just prepared grafted clones of P336 and P236 and will perform artificial pollination of the same cross. At that time, we can check the repeatability of QTLs detected in this study. However, we hesitate to describe our future plan in this manuscript.

Reviewer 3 Report

In this study, Goto and coworkers map a number of loci associated with several phenotypic traits in a fir. The study is exquisitely done. My only concern is that, if I understood correctly, every sapling sampled comes from only two trees. Besides this, I only have minor comments, regarding to errata:

l. 3: adaptation.

l. 20: The first “T” is in bold.

l. 68: populations.

l. 92-93: What do you mean with A33, A39, P236, P336?

l. 199: an extra space before CA.

l. Figure 1: the figure is difficult to see, especially due to white colour. Please, change the colour.

l. 292: The cap of the table 2 should be in the next page.

Ref 51: Populus in cursive.

Author Response

Dear Reviewer 3 

Thank you for helpful suggestions. We revised the manuscript according to the comments. 

In this study, Goto and coworkers map a number of loci associated with several phenotypic traits in a fir. The study is exquisitely done. My only concern is that, if I understood correctly, every sapling sampled comes from only two trees. Besides this, I only have minor comments, regarding to errata:

  1. 3: adaptation.

Response: We corrected this term. Thank you for your suggestion.

  1. 20: The first “T” is in bold.

Response: We corrected this. Thank you for your suggestion.

  1. 68: populations.

Response: We revised the expression and added Table 1 in the current version.

  1. 92-93: What do you mean with A33, A39, P236, P336?

Response: Thank you for your comment. We understand that information on grandparents and parents was lacking in the previous version. Thus, we added Table 1 to the current version to help readers understand the genetic composition of the segregated population.

  1. 199: an extra space before CA.

Response: Thank you for your comment. We corrected this.

  1. Figure 1: the figure is difficult to see, especially due to white colour. Please, change the colour.

Response: Thank you for your comment. We revised Figure 1 according to your comment.

  1. 292: The cap of the table 2 should be in the next page.

Response: Thank you for your comment. We rearranged the table and figures by adding the new table to the current version.

Ref 51: Populus in cursive.

Response: Thank you for your comment. We corrected this.

Round 2

Reviewer 2 Report

Dear Authors,

Thank you for your response to my comments on your ms and for including additional information both in M, R, and D sections. For sure that made the ms more clear. 

I do agree with most of your reasoning concerning association mapping analysis. Yes, usually it is not conducted on biparental populations. Nevertheless, such analysis is helpful when many markers are excluded due to missing data or segregation ratio. In your case markers with more than 20% missings were deleted, furthermore, markers with segregation distortion were also eliminated (which is normal). Thus, a lot of information that could support your mapping data, QTL analysis, etc might be missing. That is why I have suggested running association mapping (to retrieve that information). Based on my own practice I do like including such analyses as they support (complement) my QTL results if associated markers are the same as those identified in the QTL analysis (it is not always the same and in some instances, there is a shift between QTLs detected based on QTL analysis and association mapping). Anyway, the incorporation of association studies was just a suggestion. This it is your choice whether to include such data or not. 

Concerning normality. Yes, you are completely right that there is no sense to verify the normal distribution of continuous traits as indicated by the Central Limit Theorem. However, running, i.e., the Kolmogorov-Smirnov test (even if it rejects normality) is usually recommended. Maybe it is worth indicating that based on the CLT you do not need to verify normality? Again, I leave this on your side which is the best choice.

Wish you luck!